# The Deep Learning Recipe for LLM Applications

## Abstract

Large Language Models (LLMs) have revolutionized AI research and enabled exciting applications. To build a complex LLM application, such as an LLM agent, most existing research relies on insights from other domains or heuristics to manually build the application. However, this approach often requires heavy hand-engineering and fails to fully optimize for the downstream task of interest. Inspired by the tremendous success of deep learning, we proposed to construct LLM applications in a modular manner, similar to building a deep neural network. Our key insight is to make analogies between LLM building blocks, such as retrievals, memories, and prompting strategies, and the successful deep learning modules, such as MLPs, attention, and recurrent modules. We further design forward inference and feedback mechanisms for LLMs, where prompts in LLMs are considered as the weights in deep models, and the prompt optimization from feedback is analogous to the back-propagation algorithm. We additionally leverage a search algorithm to search for the best configuration of LLM applications, similar to the neural architecture search (NAS) in deep learning research. Comprehensive experimental results demonstrate that the proposed deep learning recipe for LLM applications is highly effective, in particular: (1) Organizing LLM modules into deep-learning-style architectures yields noticeable performance gain; (2) Automatic prompt optimization, equivalent to backpropagation, is efficient in incorporating feedback from the task of interest and achieves at least 5% performance improvement; (3) NAS equivalent algorithm works well for further optimizing the LLM application architecture with 11% performance gain compared with randomly designed architectures. Overall, our research demonstrates the exciting opportunity of transferring the success of deep learning to building LLM applications.

## 1 Introduction

Large Language Models (LLMs) have demonstrated remarkable potential in achieving Artificial General Intelligence (AGI) due to their impressive planning and reasoning abilities (Wu et al., 2023b; Ge et al., 2023), which has sparked an upsurge in studies investigating sophisticated LLM applications such as AI agents (Zhao et al., 2023; Wang et al., 2023; Peng et al., 2023). LLM applications usually involve various LLM building blocks (Packer et al., 2023; Yao et al., 2023; Topsakal & Akinci, 2023; Pandya & Holia, 2023; Jeong, 2023), such as retrieval (Robertson et al., 2009; Izacard et al., 2022; Lin et al., 2023), memory modules (Zhao et al., 2023; Wang et al., 2023), and prompting strategies (Yao et al., 2024). We argue that the success of LLM applications relies on the integration of multiple LLM modules (Zhao et al., 2023; Talebirad & Nadiri, 2023; Wu et al., 2023a), which still receives limited attention from the community. Therefore, our paper aims to raise attention to the pressing research question: *how to effectively and automatically construct LLM applications based on basic building blocks.*

Constructing effective LLM applications often involves leveraging insights from various disciplines, such as neuroscience (Yao et al., 2022; Shinn et al., 2023; Kwon et al., 2023) and computer architecture (Packer et al., 2023; Talebirad & Nadiri, 2023). For example, MEMGPT (Packer et al., 2023) draws inspiration from hierarchical memory systems in traditional operating systems to construct an LLM agent application with several manually designed LLM building blocks. However, applying such interdisciplinary insights to LLM application construction has the following shortcomings: (1) The approach is often domain-specific and not

generic, requiring heavy manual designs; (2) Designing the LLM application necessitates extensive trial and error, making the process both costly and inflexible.

The tremendous success of Deep Learning (DL) has brought us a new perspective on solving the above problems. In the area of DL, researchers have proposed various basic modules such as Recurrent Neural Network (RNN) (Li et al., 2019; Selvin et al., 2017; Jordan, 1997) and Attention Network (Wang et al., 2017; Fu et al., 2019; Hou et al., 2019) to process various tasks. Constructing the basic modules into complicated architectures like Transformer (Vaswani et al., 2017b; Han et al., 2021) and further leveraging the forward and back propagation (Johansson et al., 1991; Buscema, 1998) to train such architectures have further contributed to the success of DL on tackling complex real-world applications.

Inspired by the success of DL, we propose to *design LLM applications with a Deep Learning recipe.* Our approach involves *defining building blocks of LLMs analogous to the DL modules, constructing LLM applications similar to defining NN architectures, and developing the prompt optimization algorithm of LLMs similar to the backpropagation algorithm.* Specifically, we explore LLM modules including: (1) vanilla LLM as a single layer MultiLayer Perceptron (MLP) (Popescu et al., 2009); (2) Memory processer (Zhang et al., 2023; Packer et al., 2023) as an RNN; (3) Retrieval (Robertson et al., 2009; Izacard et al., 2022; Lin et al., 2023) as an attention network (Wang et al., 2017; Fu et al., 2019; Hou et al., 2019); (4) prompting strategies (Besta et al., 2023; Yao et al., 2024) as Graph Neural Networks (GNNs) (Zhang et al., 2021; Shi et al., 2019; Kenning et al., 2022). Based on the defined LLM modules, we could construct more complex LLM applications by referencing successful DL architectures, notably, Transformer (Vaswani et al., 2017a). Furthermore, we design a meta prompt module similar to the training parameters in DL to guide the forward inference, and a feedback-propagation module that incorporates the external feedback and iteratively updates the meta prompt similar to the backpropagation algorithm. We additionally leverage a search algorithm to search the best configuration of LLM applications, similar to the neural architecture search (NAS) (Liu et al., 2018) in DL research. In summary, our main contributions are as follows:

- The first work to investigate constructing LLM applications from the DL perspective and recipe, extends the science and engineering of LLM applications.

- Proposing DL-style building blocks for LLMs and constructing a "Transformer" architecture for LLM application. Observing at least 5% performance gain by "backpropagating" through the LLM application via a novel feedback-propagation propagation algorithm.

- An automatic search algorithm for the best configuration of LLM application, similar to NAS in DL, brings 11% performance gain compared to prior practice.

## 2 Related Works

In this section, we first revisit the basic DL modules and how they are used to build complex NN architectures. We then review the current approaches in building LLM applications and discuss how can we bring insights from the success of DL applications to LLM applications.

**Modules of Deep Learning.** In the area of DL, researchers have designed various basic modules to tackle different tasks. In the early period of DL design, researchers propose MLP (Popescu et al., 2009) to solve tasks such as classification, prediction, and regression. However, this basic architecture is less effective for complex input data. Consequently, researchers have developed specialized modules with various functionalities. RNNs (Li et al., 2019; Selvin et al., 2017; Jordan, 1997) are tailored for sequential data, maintaining a hidden state to capture information across time steps. CNNs (Kiranyaz et al., 2015; 2019) excel at capturing local patterns within images. Additionally, Attention mechanism (Wang et al., 2017; Fu et al., 2019; Hou et al., 2019) further refines model focus by emphasizing the most salient segments within the data. By combining and arranging these specialized modules, researchers have developed numerous powerful architectures. For instance, the Transformer (Vaswani et al., 2017b; Han et al., 2021), one of the most popular architectures today, relies on the attention mechanism and simple feed-forward layers. It has demonstrated surprising results in generative AI. Additionally, some researchers have integrated the attention mechanism into RNNs, yielding very promising outcomes (Wang & Tax, 2016; Merity, 2019). Researchers have further developed

automating architecture engineering, such as Neural Architecture Search (NAS) (Liu et al., 2018; Zoph & Le, 2016; Pham et al., 2018), aiming to find the optimized design of our machine learning model.

**Construction of LLM Applications.** Much of the current research on LLM application relies on the insights and observations from other domains (Yao et al., 2022; Shinn et al., 2023; Kwon et al., 2023; Chen et al., 2023). However, these designs are often problem-specific and do not offer a systematic approach to defining a good LLM application architecture. Recent efforts, like Langchain [1] and LlamaIndex [2], aim to offer standardized building blocks and modules for LLM applications. However, they still lack a systematic approach to designing different LLM applications for various tasks, providing limited guidance on their integration for sophisticated and meaningful architectural compositions. In response, we propose the development of standardized modules for LLM application design, inspired by the proven methodologies within the DL domain. In this work, we show that these basic building blocks can be synthesized into a coherent framework, offering a novel, modular approach to architectures in LLMs that builds a bridge between LLMs and DL.

# 3 The Deep Learning Recipe for LLM Applications

To explore the deep learning recipe for LLM applications, we first introduce the basic building blocks of LLMs in Sec 3.1, which demonstrates the relationships between DL and LLM building blocks. Inspired by the success of DL's training process, we then illustrate the forward inference and feedback propagation of LLM in Sec 3.2. Finally, we introduce the instantiations of: building a "Transformer" with LLM building blocks in Sec 3.3.

## 3.1 Deep-Learning-Style Building Blocks of LLM Applications

To introduce the basic building blocks of LLM and the relationship between DL and LLMs, we "translate" the modules of LLM into modules of DL as shown in Figure 1.

**Vanilla LLM as Single Layer MLP.** As a basic component when constructing an LLM application, an LLM receives a query $q$ and outputs a response $r$. It can be translated as a single layer MLP, as shown in Figure 1(a), which also plays a fundamental role in DL (Popescu et al., 2009).

**Memory Processer as Recurrent Neural Network.** During the interaction process of the LLM, it will store the knowledge obtained by the interaction as memories $\{m_0, m_1, ...m_n, q\}$ and utilize these memories for future response (Shinn et al., 2023; Kwon et al., 2023) as shown in Figure 1(b). Similarly, Recurrent Neural Network (RNN) excels at processing sequential data due to their inherent structure that allows for the storage and utilization of previous information (Li et al., 2019; Selvin et al., 2017; Jordan, 1997). It achieves this by maintaining hidden states $h_t$ that are updated at each time step $t$, effectively encoding the history of the sequence $\{m_0, m_1, ...m_n, q\}$ up to that point. Therefore, we regard the memory processor as RNN.

**Retrieval as Attention Network.** When LLMs interact with factual knowledge stored in the corpus context $C$, they utilize retrieval (Robertson et al., 2009; Izacard et al., 2022; Lin et al., 2023) to efficiently retrieve accurate context for future response. As shown in the left part of Figure 1(c), to retrieve the accurate context, retrieval exploits the embedding similarity between query $q$ and corpus context $C$ and obtains the retrieved context $C_r$ of the top $K$ similarity ranking. This process is similar to the attention network shown in the right part of Figure 1(c). They both utilize the embedding similarity between query $q$ and corpus context $C$ to obtain an important score $f$, which determines the important weights of each context. From this perspective, we translate the retrieval as an attention network. To be specific, single-query retrieval can be interpreted as single-head attention and multi-query retrieval can be regarded as multi-head attention (Wang et al., 2017; Fu et al., 2019; Hou et al., 2019).

**Prompting Strategies as Graph Neural Network.** As shown in Figure 1(d), common prompting strategies, such as Tree of Thoughts (ToT) (Yao et al., 2024) and Graph of Thoughts (GoT) (Besta et al., 2023) both decompose query $q$ into interconnected thoughts, which are connected and related to each other.

---

[1] https://www.langchain.com
[2] https://www.llamaindex.ai

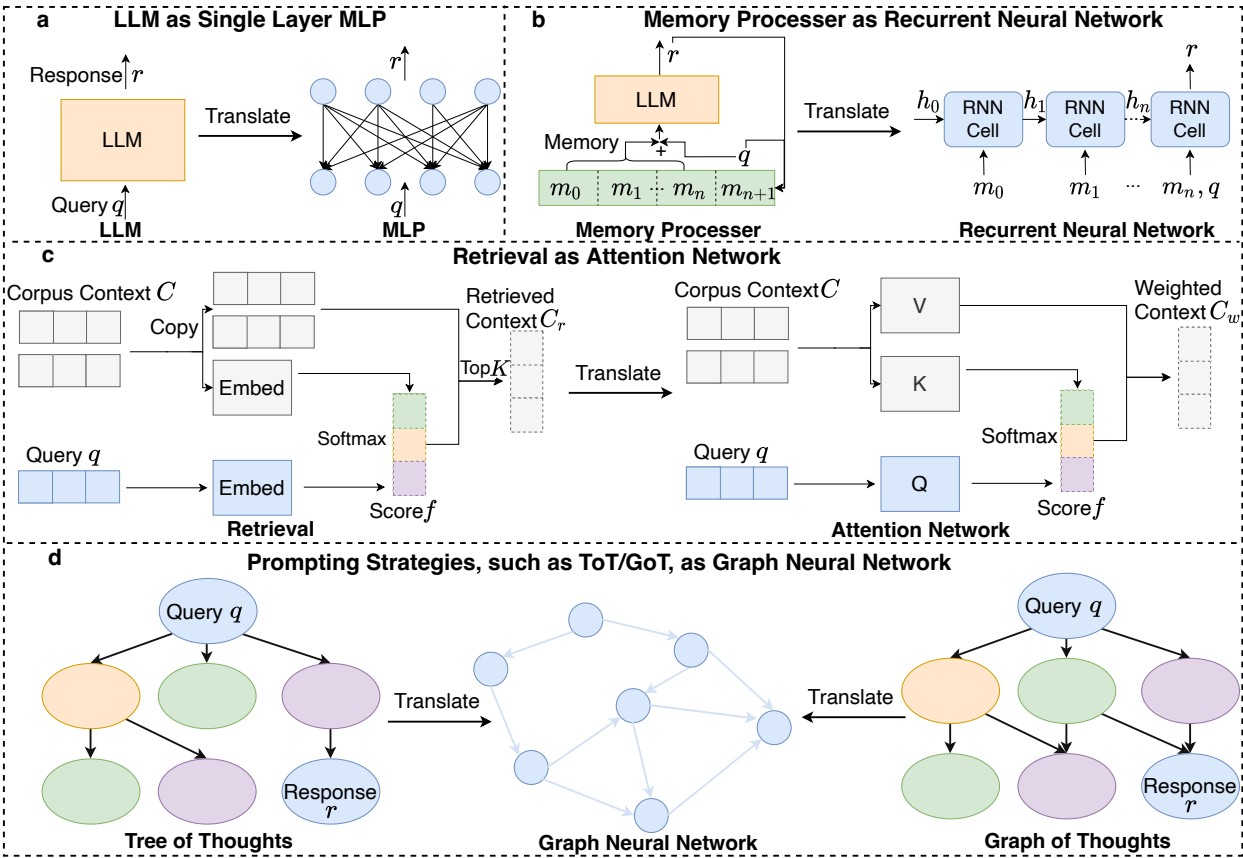

Figure 1: **Basic Building Blocks of LLM applications**. **(a)** A vanilla LLM can be translated into a single layer MLP, which receives a query $q$ and outputs a response $r$; **(b)** Memory processor can be regarded as RNN since they both utilize historical data for future planning; **(c)** Retrieval is analogous to the attention network of DL because they both extract the importance information of the corpus context; **(d)** Prompting strategies, such as Tree/Graph of thoughts, are translated to GNNs since they decompose query $q$ into interconnected thoughts, which are connected and related to each other.

These query and thought nodes and their relationships can be abstracted into nodes and edges of Graph Neural Networks respectively. For convenience, we translate ToT and GoT into TNN and GNN in DL respectively in the subsequent introduction.

## 3.2 Forward Inference and Feedback Propagation for LLM Applications

The success of DL not only lies in the design of fundamental modules discussed in Sec 3.1, but also in the training process containing forward and back propagation (Buscema, 1998) that brings external signals and feedback to the DL application. Inspired by this, we design the forward inference and feedback propagation of LLMs as shown in Figure 2.

**Parameters of the LLM Application.** The parameters can be divided into two categories: training parameters and hyperparameters. Here, the meta prompt in Figure 2 is equivalent to training parameters in deep learning, since it guides forward inference and can be updated through feedback propagation. Moreover, hyperparameters in LLM applications include max tokens of each LLM (size of MLP); number of retrievals (head number of attention network); retrieve number of each retrieval (mask); node number or depth of tree/graph of thoughts (node number or layer number of GNN); etc. These are predefined before using the LLM applications.

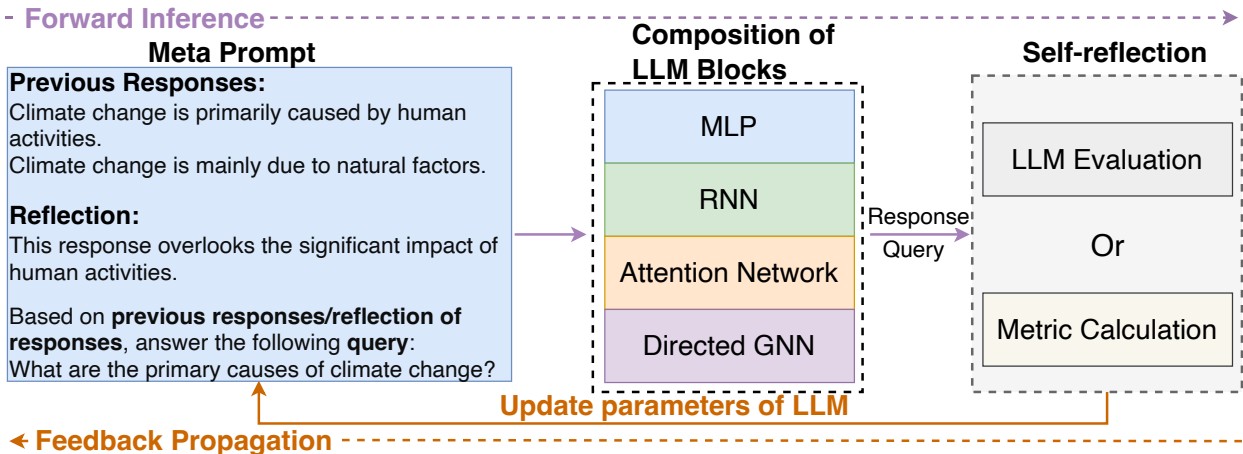

Figure 2: **Forward and Back Propagation Algorithms for Large Language Model**. In this process, meta prompt is regarded as trainable parameters in DL. For the forward inference, the input query goes through the composition of LLM building blocks to obtain the response, which is guided through the meta prompt. For the feedback propagation, the feedback from the self-reflection updates the parameters (meta prompt) of LLM.

**Forward Inference.** In the forward inference process of DL, the input data passes through each block of the neural network to obtain the final output (Buscema, 1998; Prabhushankar & AlRegib, 2022). Similar to this, in the forward inference process of LLM, the input query goes through the composition of LLM building blocks to obtain the response as shown in Figure 2. Here we set up a meta prompt to guide the forward inference of LLM, which contains the input query, previous responses, and reflection returned by the self-reflection module.

**Feedback Propagation.** Many researchers contribute the success of DL to the feedback propagation process of DL (Buscema, 1998; Luft, 2014), which updates the parameters of DL through feedback of losses between outputs and ground truths. Inspired by this, as illustrated in Figure 2, we regard the meta prompt as parameters of LLM and design a self-reflection module as the loss function to evaluate the effect of response on input query and update the parameters of LLM. Specifically, the self-reflection contains LLM Evaluation and Metric Calculation. For some queries whose response quality has clear metrics, we will choose Metric Calculation. Otherwise, we utilize LLM Evaluation to make LLM judge the quality of the response. The generated reflections and responses will be used to update the meta prompt to guide the LLM response of the next round.

### 3.3 Case study: Building a "Transformer" with LLM Building Blocks

In this section, we introduce how to build a (cross-attention) Transformer with LLM building blocks. As illustrated in Figure 3, we show the correspondence between the main modules of the cross-attention transformer and LLM building blocks to illustrate how to utilize LLM building blocks to build a transformer. For the cross-attention transformer, the two different features of data will first go through a cross-attention network and the output will be obtained through an MLP layer. To avoid the vanishing gradient problem or gradient explosion, researchers will also add the residual module to improve the training performance of the transformer (Vaswani et al., 2017b; Han et al., 2021). The success of the transformer in integrating and processing information from different features has inspired the design of LLM. As shown in the right part of Figure 3, we analogize the raw query and the corpus context into two different features of data, like $Feature_1$ and $Feature_2$. On this basis, we first design a meta prompt containing raw queries to guide the LLM to generate multiple queries. Then we propose a multi-query retrieval based on multiple queries to retrieve relevant context from corpus context, which is equivalent to the multi-head cross-attention network. Additionally, we add the raw query to the generated multiple queries to ensure the effectiveness of our LLM, which is similar to the design of the residual module. Finally, an LLM will output the response like an MLP

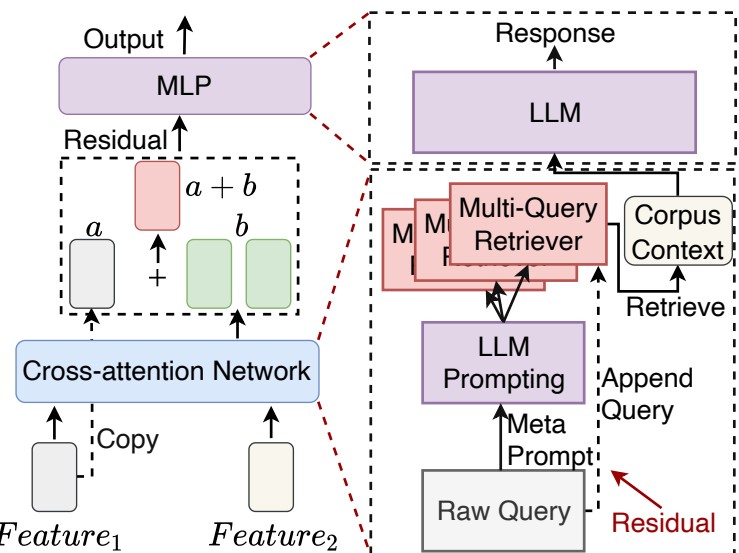

Figure 3: **Build a Cross-attention Transformer with LLM Building Blocks**. The multi-query retrieval is equivalent to the cross-attention network and the LLM is translated to MLP in DL. Additionally, the process of appending the raw query to the generated multiple queries is equivalent to the residual.

based on the retrieved context and input query. We summarize the DL recipe for LLM applications as a user instruction manual in Table 5 and please refer to Appendix A for details.

## 4 Experimental Setup

### 4.1 Datasets and Tasks

We evaluate LLM applications on two major categories of tasks. They differ in whether the LLMs interact with external knowledge (Sun et al., 2023), such as external factual knowledge.

**LLM without External Knowledge.** We evaluate non-retrieval-based methods using two datasets: (1)**Traveling Salesman Problem (TSP)** (Hahsler & Hornik, 2007), where we create instances by randomly generating $n$ nodes with $x$

Table 1: **Overview of Datasets.**

| Dataset | Task Type | Cases |
|---|---|---|
| **Without External Knowledge** | | |
| TSP | Optimization | 30 |
| GSM8K | Math Problem | 30 |
| **With External Knowledge** | | |
| StrategyQA | Commonsense Reasoning | 30 |
| Related-multi | Summary | 30 |

and $y$ coordinates in the $[-10, 10]$ range. As a classic combinatorial optimization problem, performance is evaluated by the length of the best solution within the fixed exploration step. (2) **GSM8k** (Cobbe et al., 2021), which contains high-quality, linguistically diverse grade-school math word problems necessitating multi-step reasoning, crafted by expert problem writers. Each problem includes a detailed reasoning process and the correct final answer, with performance evaluated based on final answer accuracy.

**LLM with External Knowledge.** Our evaluation of retrieval-based frameworks utilizes two datasets: (1) **StrategyQA** (Geva et al., 2021), a benchmark for open-domain commonsense reasoning question-answering that highlights the importance of implicit reasoning. It involves retrieving pertinent evidence paragraphs from a corpus to correctly answer questions. We compare the retrieved paragraphs with the ground truth and obtain Precision, Recall, and F1 as evaluation metrics. (2) **Related-multi** (will be released after review), designed to assess a language model's capability to effectively use extensive context from real-world academic papers for answering queries. Specifically, the task is to write a related work section based on the title and

Given a simple mathematical question along with any previous attempts to solve it, please directly provide the final answer. Be aware that the details about past explorations may contain inaccuracies.

Question: {question};
Previous Responses: Here are the previous responses, which may not be accurate: {pre_res};
Evaluation for the Past Response: {eval}

Your response should follow the structure outlined below:
R: <Replace Here With Your Reasonings>;
A: Place your Final Answer here as a clear numeric value. Ensure there are no additional words, signs, or explanations! Enclose the numeric value in angle brackets.

An example of the desired output is:
R: First find the total number of starfish arms: 7 starfish * 5 arms/starfish = <<7*5=35>> arms \n Then add the number of seastar arms to find the total number of arms: 35 arms + 14 arms = <<35+14 = 49>> arms\n
A: <49>

Figure 4: An example of the meta prompt used for RNN-MLP on the GSM8K dataset. The blue text contains input questions and previous responses/evaluations. The purple text describes the output format instructions.

abstract of a target paper. LLM needs to use the title and abstract as the query to retrieve text chunks to complete this task. Text chunks depict the abstracts of several papers (each text chunk corresponds to the abstract of a paper), where some papers are cited in the related work section of the target paper, while others are randomly sampled from the same broader field. The retrieved text chunks are compared with the ground truth to obtain Precision, Recall, and F1 as evaluation metrics.

## 4.2 Models: Various Architectures for LLM Applications

We delve into the detailed descriptions of the architectures we explored, with a particular emphasis on their implementations and the methods employed for backward propagation across various modules. For the two LLM datasets without external knowledge, we design the following four basic architectures: (1) **MLP:** a single LLM; (2) **RNN-MLP:** previous responses are appended into the meta prompt to guide the generation of response; (3) **TNN-MLP:** utilize ToT to generate multiple thoughts for response generation; (4) **GNN-MLP:** GoT is exploited to develop various thoughts for response generation.

We also develop four basic architectures for LLM applications with external knowledge. In order to interact with external knowledge, the four architectures are supplemented with the attention to retrieve relevant information based on the above architectures: (1) **Att-MLP:** a single retrieval to retrieve relevant information for LLM; (2) **RNN-Att-MLP:** previous responses are appended into the meta prompt to change the query for retrieval; (3) **TNN-Att-MLP:** utilize ToT to generate new query for retrieval; (4) **GNN-Att-MLP:** GoT is exploited to develop new query to retrieve. More details about the basic architectures of LLM applications are introduced in Appendix C. Additionally, we have shown an example of the meta prompt used for RNN-MLP on the GSM8K dataset in Figure 4. Other meta prompts are also summarized in Appendix B.

## 5 Results

### 5.1 Performance of Different Architectures of LLM Applications

**Different Applications Lead to Different Ideal LLM Architectures.** Our results across two distinct types of datasets are illustrated in Figures 2 and 3, demonstrating that the integration of RNN, TNN, and GNN generally enhances the MLP's performance across various tasks. Specifically, in reasoning (GSM8K) and optimization (TSP) problems that do not require external knowledge, GNN, and TNN excel due to their strong logical capabilities. For tasks involving reasoning (StrategyQA) and long-context summary (Related-multi) that necessitate external knowledge, RNNs proved most effective, in line with our expectations. This effectiveness is attributed to their capability to process and retain long sequences of information. Consistent with findings in DL, combining RNNs with attention mechanisms has been shown to yield promising results.

Table 2: **Performance of Different LLM applications on TSP and GSM8k**.

| Type | Without External Knowledge | |
|---|---|---|
| Dataset | TSP | GSM8k |
| Method | Length | Accuracy |
| MLP | 79.2 | 53.3% |
| RNN-MLP | 76.6 | 60.0% |
| TNN-MLP | 73.2 | 66.7% |
| GNN-MLP | 73.9 | 64.3% |

Table 3: **Performance of Different LLM applications on StrategeyQA and Related-multi**.

| Type | With External Knowledge | | | | | |
|---|---|---|---|---|---|---|
| Dataset | StrategeyQA | | | Related-multi | | |
| Method | Precision | Recall | F1 | Precision | Recall | F1 |
| Att-MLP | 0.45 | 0.40 | 0.42 | 0.31 | 0.17 | 0.22 |
| RNN-Att-MLP | 0.72 | 0.62 | 0.66 | 0.33 | 0.19 | 0.24 |
| TNN-Att-MLP | 0.53 | 0.47 | 0.50 | 0.34 | 0.20 | 0.25 |
| GNN-Att-MLP | 0.58 | 0.51 | 0.54 | 0.29 | 0.16 | 0.21 |

**Feedback-Propagation Effectively Optimizes of LLM Application Performance without Human Intervention.** Through our experiments, we discover that our automatic prompt optimization technique, akin to backward propagation, consistently improves the performance of the LLM applications in most scenarios. This verifies that the LLM applications benefit from integrating past experiences and self-evaluations. Figure 6 presents the results at different steps on the TSP datasets, illustrating that feedback propagation can bring at least 5% performance gain for LLM applications. Moreover, we can also observe that GNN-MLP and TNN-MLP iterate faster than other architectures and obtain better results. This is because they explore more solutions at each step and improve the efficiency of optimization.

### 5.2 Case Study: Results of "Transformer-style" LLM Applications

We utilize variants of the transformer as an example to show the role of different LLM building blocks when building complex architectures. To be specific, we set up the following variants. (1) **Transformer**: It denotes transformer illustrated in Sec 3.3. (2) **RNN-Transformer**: We add previous responses to the meta prompt of the transformer to utilize its memory, which is commonly used in DL (Xia et al., 2019; Liu et al., 2019) when processing historical data. (3) **TNN-Transformer/GNN-Transformer**: ToT/GoT is added to transformer to guide multi-query generation, which is similar to GraphTransformer (Yun et al., 2019; Hu et al., 2020; Rampášek et al., 2022) in DL. We compare their performance on two datasets with the same hyperparameters and report all the results in Figure 7. It can be observed that the Transformer performs best in StrategyQA. This is because StrategyQA's tasks are relatively simple, and Transformer is already relatively complex, therefore other variants do not perform well. We can also observe that TNN-Transformer and GNN-Transformer perform relatively better

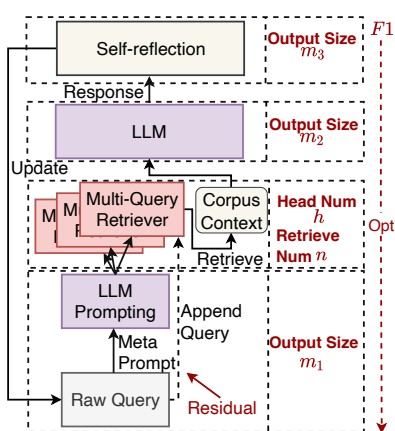

Figure 5: **Optimization on transformer to maximize F1.**

than other variants on Related-multi. It is because writing-related work requires more thinking and divergence of thoughts (Randolph, 2019; Torraco, 2005).

### 5.3 Automatic Architecture Search of LLM Applications

To automatically optimize the LLM application architecture like NAS, we propose an optimization framework based on Optuna (Akiba et al., 2019), which is an efficient and lightweight hyperparameter optimization software. To better illustrate our framework, we take the optimization of the transformer introduced in Sec 3.3 on Related-multi as an example. As shown in Figure 5, we optimize the hyperparameters of each module using Optuna. To be specific, we first conclude the hyperparameters and define their optimization range in

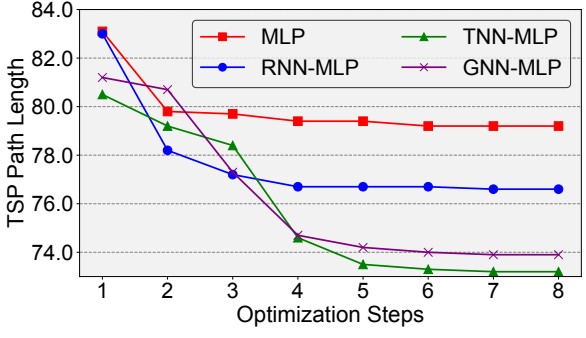

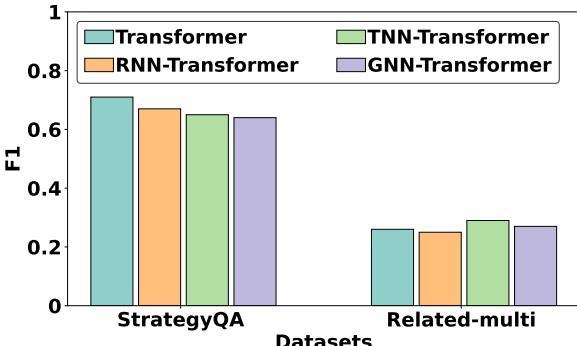

Figure 6: **The performance of LLM applications on the TSP dataset will get better as the feedback propagation progresses.**

Figure 7: **Case Study: Using LLM Building Blocks to Build Variants of Transformer**.

Table 6 of Appendix D: Output size $m_1$ (max tokens) for LLM prompting, head num $h$ and retrieve num $n$ of each head for multi-retrieval, output size $m_2$ for output LLM, and output size $m_3$ for self-reflection. We utilize 30% of the dataset (training set) for searching optimization architecture and 70% for testing (testing set) and further define F1 as the optimization goal. For each set of hyperparameters $pr_i = (m_1, m_2, m_3, h, n)$, we calculate its F1 performance on training set. As illustrated in Figure 16 of Appendix D, we can obtain $\{pr_i, F1_i\}^k$ via $k$ trials (120 trials in our experiment) and optimize the architecture by maximizing F1. In addition, as shown in Table 4, we select some points to illustrate that the LLM application becomes better as the number of trials increases. Through the experiment, we obtain the optimized hyperparameters $pr = (m_1 = 192, m_2 = 192, m_3 = 320, h = 2, n = 4)$ with $F1 = 0.712$ on the testing set, which has 11% better performance than randomly designed architectures (obtained by conducting experiments in which $m$ groups of hyperparameters were randomly selected and averaging the corresponding F1 scores).

## 6    Conclusion and Discussion

**Conclusion.**   In conclusion, our work introduces a pioneering modular approach to the construction of LLM applications, establishing parallels with DL. We provide comprehensive methods and insights into viewing and assembling LLM building blocks in a manner analogous to DL architectures. Through extensive experimentation, we demonstrate that specific classic LLM modules are

Table 4: **As the number of LLM architecture searches increases, the LLM application is getting better.**

|  | 0 | 7 | 43 | 78 |
|---|---|---|---|---|
| F1 Score | 0.541 | 0.685 | 0.694 | 0.712 |

optimized for distinct tasks, similar to those in DL, and can be integrated in a manner reminiscent of DL configurations. Moreover, our feedback system enhances the performance of LLMs, paralleling the training mechanisms in DL. Finally, we propose an automatic optimization method for identifying the most effective LLM frames, presenting a promising avenue for future research.

**Discussion.**   This work represents a pioneering effort in the development of systematic and scientifically grounded LLM applications, drawing upon DL insights to inaugurate a new era of efficiency and standardization in the research and application of LLMs. By establishing fundamental LLM applications, our objective is to reconcile the existing diversity of paradigms within LLM research and, akin to DL, introduce classic and standardized modules that enhance scientific rigor, effectiveness, and innovation within the academic community. In the industrial context, the adoption of foundational and systematic architectures enables the creation of more efficient LLM building blocks, thus establishing a solid basis for the field of LLM applications. These advanced LLM building blocks possess the potential to streamline the deployment of LLM products, thereby enabling significant real-world impact and facilitating transformative changes across various sectors. Inspired by the progression of DL, this work introduces a suite of standardized modules within the LLM application, setting a foundation for future research to explore multi-LLM settings, integrate more established

DL modules and architectures like CNN, and adopt self-exploration algorithms akin to NAS to revolutionize the optimization of LLM applications, potentially accelerating the discovery of innovative solutions.

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

## A  User Instruction Manual

Table 5: Deep Learning Recipe for LLM Applications.

| LLM Applications | Deep Learning Architecture |
| --- | --- |
| Vanilla LLM | Single layer MLP |
| Memory processor | Recurrent neural network |
| Retrieval | Attention Network |
| Prompting strategies (ToT & GoT) | Graph neural network |
| Prompts | Training parameters |
| Max tokens of each LLM | Size of MLP |
| Number of retrievals | Head number of attention network |
| Node number or depth of tree/graph | Node number or layer number of GNN |
| Forward inference of LLMs | Forward inference |
| Feedback propagation | Back propagation |
| Self-reflection | Loss function |
| Retrieve number of each retrieval | Mask |

## B  Prompts

You are given a list of points with coordinates below: {question}.
Below are the previous traces and their lengths. Lower length is better.
Previous Solutions: {pre_res};

Give me a new trace that is different from all traces above, and has a length lower than any of the above. The trace should traverse all points exactly once. The trace should start with <trace> and end with </trace>.

Here is an example of the output format: '<trace> 0,3,2,5,4,7,8,1,9,6 </trace> length: 2254'. Please don't include any extra words, don't tell me your reasoning process.

Figure 8: An example of the meta prompt used for RNN-MLP on the TSP dataset. The blue text contains input questions and previous responses/evaluations. The purple text describes the output format instructions.

**ToT/GoT Strategy:**
You are given a list of points with coordinates below:
Question: {question};
Below are the previous traces and their lengths. Lower length is better.
Previous Solutions: {past_stra};
Evaluation for the Past Strategy: {eval}

Give me a new trace that is different from all traces above, and has a length lower than any of the above. The trace should traverse all points exactly once. The trace should start with <trace> and end with </trace>.

Here is an example of the output format: '<trace> 0,3,2,5,4,7,8,1,9,6 </trace> length: 2254'.

- - - - - - - - - - - - - - - - - - - - - - - - - - - - - - - - - - - - - - - - - - - - - - - - - - - - - - - - - - - -

**ToT/GoT Vote:**
Given a TSP question and 5 proposed strategies, please determine the most promising strategy as your output without additional words.
Question:{question};
Strategies: {strategies};
Please output use the following format:
<trace> PONTS </trace>.
Replace PONTS with points 1 to 9.

Figure 9: An example of the meta prompt used for TNN-MLP/GNN-MLP on the TSP dataset. The blue text contains input questions and previous responses/evaluations. The purple text describes the output format instructions.

Given a simple mathematical question along with any previous attempts to solve it, please directly provide the final answer. Be aware that the details about past explorations may contain inaccuracies.

Question: {question};
Previous Responses:  Here are the previous responses, which may not be accurate: {pre_res};
Evaluation for the Past Response: {eval}

Your response should follow the structure outlined below:
R: <Replace Here With Your Reasonings>;
A: Place your Final Answer here as a clear numeric value. Ensure there are no additional words, signs, or explanations! Enclose the numeric value in angle brackets.

An example of the desired output is:
R: First find the total number of starfish arms: 7 starfish * 5 arms/starfish = <<7*5=35>> arms \n Then add the number of seastar arms to find the total number of arms: 35 arms + 14 arms = <<35+14 = 49>> arms\n
A: <49>

Figure 10: An example of the meta prompt used for RNN-MLP on GSM8K dataset. The blue text contains input questions and previous responses/evaluations. The purple text describes the output format instructions.

---

**ToT/GoT Strategy:**
Given a question and past strategy with evaluations, your task is to provide 5 different potential strategies to solve it.
Question: {question};
Previous Strategy: {past_stra};
Evaluation for the Past Strategy: {eval}

Please format your response by listing the strategies separately.

Here is an example:
Original Question: Every tree that Bart cuts down gives him 75 pieces of firewood. If he burns 5 logs a day from November 1 through February 28, how many trees will he need to cut down?
Strategy: November has 30 days, December has 31 days, January has 31 days and February has 28 days for a total of 30+31+31+28 = <<30+31+31+28=120>>120 days\nHe burns 5 pieces of wood every day so 120*5 = <<120*5=600>>600 pieces of wood\nEvery tree he cuts down supplies 75 pieces of firewood and he will burn 600 pieces so he needs 600/75 = <<600/75=8>>8 trees

- - - - - - - - - - - - - - - - - - - - - - - - - - - - - - - - - - - - - - - - - - - - - - - - - - - - - - - -

**ToT/GoT Vote:**
Given a question and 5 proposed strategies, please determine the most promising strategy as your output.
Question:{question};
Strategies: {strategies};
Please output use the following format:
To solve the question, we should <Content Of The Best Strategy>

Figure 11: An example of the meta prompt used for TNN-MLP/GNN-MLP on GSM8K dataset. The blue text contains input questions and previous responses/evaluations. The purple text describes the output format instructions.

---

Given an original question and the past responses, your task is to enhance the original question for better retrieval outcomes.

Question: {question};
Previous Resposnes for references: {past_stra};
Evaluation on the Last Response: {past_eval};

Here is an example:
Original Question: Is Christmas celebrated during winter?
Response: What is the date of Christmas, and does it occur in winter?

Figure 12: An example of the meta prompt used for RNN-Att-MLP on StrategeyQA dataset. The blue text contains input questions and previous responses/evaluations. The purple text describes the output format instructions.

**ToT/GoT Strategy:**
Given a question and past strategy with evaluations, your task is to provide 5 different potential strategies to solve it:
Question: {question};
Previous Solutions: {past_stra};
Evaluation for the Past Strategy: {eval}

Please format your response by listing the strategies separately.

Here is an example:
Original Question: Are more people today related to Genghis Khan than Julius Caesar?
Identified Questions: 1. 'Firstly, we need to investigate the number of kids Julius Caesar have, then we need to investigate the number of kids of Genghis Khan, then we do comparison to determine which number is greater.', ...

- - - - - - - - - - - - - - - - - - - - - - - - - - - - - - - - - - - - - - - - - - - - - - - - - - - - - - -

**ToT/GoT Vote:**
Given a TSP question and 5 proposed strategies, please determine the most promising strategy as your output without additional words.
Question:{question};
Strategies: {strategies};
Please output use the following format:
<trace> PONTS </trace>.
Replace PONTS with points 1 to 9.

Figure 13: An example of the meta prompt used for TNN-Att-MLP/GNN-Att-MLP on StrategeyQA dataset. The blue text contains input questions and previous responses/evaluations. The purple text describes the output format instructions.

Your task is to analyze a given question, a previous answer, and the evaluation of that answer, which includes suggested perspectives. Based on this information, decompose the original query into new separate question. Each of these questions should target a critical perspective or element that is necessary for fully answering the original question, taking into account the provided evaluation. Aim to cover various aspects and considerations vital for crafting a comprehensive response.

Here are the details you will work with:
Question: {question};
Previous Resposnes for references: {past_stra};
Evaluation on the Last Response: {past_eval};

Please list the five decomposed questions separately, ensuring that each one addresses a unique and significant perspective related to the original question. Format your response by numbering each decomposed question, as shown in the example.

Here is an example:
Original Question: Will Ronda Rousey hypothetically defeat X-Men's Colossus in a fight?
Response: What is Ronda Rousey's background in combat sports?

Figure 14: An example of the meta prompt used for RNN-Att-MLP on Related-multi dataset. The blue text contains input questions and previous responses/evaluations. The purple text describes the output format instructions.

**ToT/GoT Strategy:**
Given a question and past responses with evaluations, your task is to provide 5 different potential responses to solve it:
Question: {question};
Previous Solutions: {past_stra};
Evaluation for the Past Strategy: {eval}

Please format your response by listing the strategies separately.

**ToT/GoT Vote:**
Given a TSP question and 5 proposed responses, please determine the most promising responses as your output without additional words.
Question:{question};
Responses: {responses};
Please output use the following format:
To solve the question, we should <Content Of The Best Strategy>

Figure 15: An example of the meta prompt used for TNN-Att-MLP/GNN-Att-MLP on Related-multi dataset. The blue text contains input questions and previous responses/evaluations. The purple text describes the output format instructions.

## C  Details of Architectures for LLM Applications

**MLP.**  For the single-layer MLP setup, as previously defined, the LLM processes an input query $q$ and generates a corresponding output response $r$. Subsequently, through a process of backpropagation, the LLM undertakes a self-evaluation mechanism. The outcomes of this self-evaluation $e$ and response $r$ are then integrated into the original $q$, facilitating an updated $r$.

**RNN-MLP.**  In this RNN+MLP framework, the model processes a query $q$ and a sequence of past experiences $m_0, m_1, \ldots, m_n$ with evaluations $e_0, e_1, \ldots, e_n$. At each step, the latest three experiences $m_{n-2}, m_{n-1}, m_n$ and evaluations $e_{n-2}, e_{n-1}, e_n$ are integrated into the query $q$. This enriched query is then utilized to generate the next response $m_{n+1}$. After that, we evaluate it to get $e_{n+1}$. $m_{n+1}$ and $e_{n+1}$ are then fed back through backward propagation for subsequent iterations. Figure 4 shows the example meta prompt for how RNN+MLP works on GSM8K.

**TNN-MLP.**  In this setting, we use zero-shot ToT. Specifically, a query $q$ and a meta prompt $q_m$ for decomposing $q$ into interconnected thoughts are used to construct a comprehensive prompt $p$. This prompt serves as input for an LLM, yielding a response $r$. During backward propagation, a self-evaluation is performed, and the resulting evaluation $e$ is utilized to refine $q_m$ into an updated version $q'_m$, leading to the formation of an updated prompt $p'$.

**GNN-MLP.**  This is similar to that of ToT+MLP architecture with the key difference that we allow for deeper graph-based thoughts.

**Att-MLP.**  In our single retrieval Att-MLP model, an initial input query $q$ and a meta query $q_m$ are used to generate an enhanced query $p$. Information $Cr$ is retrieved from context $C$ based on $p$, and both $q$ and $Cr$ are inputted into an LLM (MLP) to generate a response $r$. A feedback system evaluates this response, producing an evaluation $e$. Both $r$ and $e$ are then fed to $q_m$ which is then utilized to update $p$.

**RNN-Att-MLP.**  In single retrieval with RNN, the model processes an input query $q$, a meta query $q_m$, and a sequence of past experiences $m_0, m_1, \ldots, m_n$ alongside their evaluations $e_0, e_1, \ldots, e_n$. At each step, the latest three experiences $m_{n-2}, m_{n-1}, m_n$ and their evaluations $e_{n-2}, e_{n-1}, e_n$, combined with the input query $q$, are integrated into the meta query $q_m$ to generate an enhanced query $p$. Information $Cr$ is then retrieved from context $C$ based on $p$, and both $q$ and $Cr$ are fed into an LLM (MLP) to produce a response $m_{n+1}$. This response is evaluated by a feedback system, generating an evaluation $e_{n+1}$. Both $m_{n+1}$ and $e_{n+1}$ are stored for future use.

**TNN-Att-MLP.**  In the ToT-Att-MLP configuration implements zero-shot ToT before the attention module, a query $q$, a meta prompt $q_m$, and a meta query $q_n$ are provided. Initially, $q$ is processed through $q_m$ to decompose into interconnected thoughts $T1, .., T_n$. Then we vote for the best thought as our strategy$s$. This strategy $s$ along with $q$ are inputted into $q_n$ to produce an enhanced query $p$. Based on $p$, information $Cr$ is retrieved from the context $C$, and both $q$ and $Cr$ are fed into an LLM (MLP) to generate a response $r$. A feedback system then evaluates $r$, yielding an evaluation $e$. This evaluation $e$ is subsequently fed to $q_m$ for updated strategy in the next iteration.

**GNN-Att-MLP.**  The GNN-Att-MLP setup is similar to the TNN-Att-MLP configuration but introduces the capability to develop deeper graph-based strategies based on previous ones. Initially, for a given query $q$, we generate a series of thoughts $T_1, \ldots, T_n$ and identify the best one among them, denoted as $T_b$. Subsequently, we construct a second layer of thoughts $T'_1, \ldots, T'_n$ based on the initial set $T_1, \ldots, T_n$. The most effective strategy $s$ is then selected from among $T_b$ and $T'_1, \ldots, T'_n$. This chosen strategy $s$ undergoes the same procedure as described in the TNN-Att-MLP setting.

## D  Automatic Optimization

Table 6: Optimization range of each hyperparameter.

|  | **Range** |
|---|---|
| Output Size $m_1$ | [64, 128, 192, 256, 320] |
| Output Size $m_2$ | [64, 128, 192, 256, 320] |
| Output Size $m_3$ | [64, 128, 192, 256, 320] |
| Head Num $h$ | [1, 2, 3, 4, 5] |
| Retrieve Num $n$ | [1, 2, 3, 4, 5] |

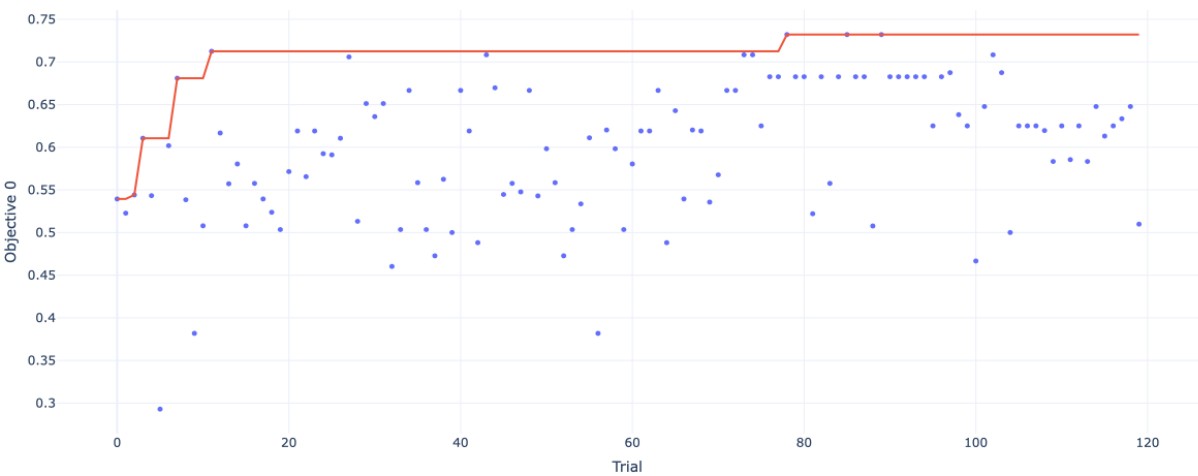

Figure 16: **The optimization process on transformer**. The x-axis represents the number of trials. For each trial, we will get a set of $pr, F1$; the y-axis represents the $F1$ value.

