# OpenReview forum: "The Deep Learning Recipe for LLM Applications"
_TMLR — Rejected by TMLR_

### Review · Reviewer_Rvc3 · 2024-12-06

**Summary Of Contributions:**

This paper proposes a novel framework for constructing and optimizing LLM applications by drawing parallels with deep learning architectures and training paradigms. The authors introduce a systematic approach to mapping LLM components to deep learning modules, where vanilla LLM corresponds to single layer MLP, memory processing maps to RNN, retrieval mirrors attention networks, and prompting strategies align with graph neural networks. They develop a "feedback propagation" mechanism for LLMs that mimics backpropagation in neural networks, enabling automated optimization of prompts and architectures. The work also introduces a search algorithm for finding optimal LLM application architectures, similar to NAS. Through experimental validation, they demonstrate significant performance improvements, achieving 5% gains through feedback propagation optimization and 11% improvement through architecture search compared to random designs. This work represents a significant step toward standardizing and automating the development of LLM applications by leveraging established deep learning principles.

**Audience:**

Yes

**Broader Impact Concerns:**

N/A.

**Claims And Evidence:**

No

**Requested Changes:**

I think the paper requires expanded experimental validation including the full GSM8K dataset (8,792 samples) rather than just 30 cases, evaluation on harder benchmarks like MATH, MMLU, or GPQA, and statistical significance analysis of results. The theoretical foundations need strengthening with deeper analysis of architectural choices and limitations, supported by ablation studies.

The paper would be further strengthened by revising terminology to use more descriptive names that don't rely on deep learning terminology without proper justification. Adding case studies showing how the deep learning analogies led to improved designs would be valuable.

**Strengths And Weaknesses:**

## Strengths:

- Novel conceptual framework linking LLM application design to established deep learning principles
- Clear methodology for automated optimization of LLM applications
- Promising experimental results showing meaningful performance gains
- Well-structured presentation of ideas and comprehensive technical details

## Weaknesses:

### experimental validation:

- Only 30 cases per dataset, which is insufficient for robust conclusions, it may cause the experiment results cannot support the claim this paper made
- Small number of datasets (only 4) tested
- No evaluation on more challenging benchmarks like MATH or MMLU


### theoretical justification:

- The mapping between LLM components and deep learning architectures feels somewhat arbitrary
- Limited explanation of why specific neural architectures (RNN, GNN, etc.) were chosen as analogues
- Insufficient analysis of the limitations of these mappings


### Nomenclature issues:

- The use of deep learning terminology (RNN, TNN, GNN) feels forced without stronger theoretical connections
- Could cause confusion in the field by overloading established terms

---

### Review · Reviewer_XNVg · 2024-12-11

**Summary Of Contributions:**

- Identifies a pain point in deep learning: building LLM-based pipelines requires some human discretion in assembling different components.
 - Suggests a method of solving the problem, via a NAS-like meta optimization.
 - Some experiments to validate.

**Audience:**

Yes

**Broader Impact Concerns:**

no concerns

**Claims And Evidence:**

No

**Requested Changes:**

The entire paper has a detached, high-level philosophising tone. Would want to see a clear, well-motivated problem identified, and a detailed, focus solutions validated. Less "case studies", "illustrations", "akin to", and generic arguments and citations.

The communication is not effective. It's difficult to pin down why exactly. But one starting point is the strange dichotomization between LLMs and DL, like somehow LLMs are not deep learning model. Another is claiming that building LLM-based agents is a fragile and hacky process with ad hoc tuning and black magic required for success. A miracle of deep learning that essentially the same architectures and largely the same training recipes have continued to power successive generations of LLMs. It's incredible how well simplistic post-training on relatively small finetuning datasets works for alignment and the delivery of practically useful systems. Building LLM agents involves some experimentation, but generally it is principled and with strong interpretation and regularity with all aspects having been extensively studied and documented by many previous researchers.

The NAS-like approach in the paper reflects this, with the five searched dimensions all having clear standalone interpretations. Which speaks to another weakness of the paper: the first part of the paper appears to promise some general and flexible approach to composing primitive building blocks into wholly novel architectures, e.g. like how "Symbolic Discovery of Optimization Algorithms" by Chen discovers a new type of optimization step. But what it actually does appears to be using a standard hyperparameter optimization software to search over a small number of size variables, exactly the sort of pedestrian experiments that would be performed in the development of a fairly ordinary LLM-based project. It's just not a very well-posed problem because left unstated are so many other aspects of the modelling: the computational budget, dataset size, the domain, etc. We know that for enough data and/or compute larger models are almost surely better, thus the experimental results do not generalize at all.


It's tough to critique the empirical results because of the lack of detail throughout, but here's some observations:
  - The numbers quoted have almost no interpretation (a 5% improvement to what? an 11% performance gain compared with _randomly designed architectures_?). There's no attempt to connect or compare your results to competing methods from previous work?
  - What are the columns on table 4?
  - Why can't you release the "Related-Multi" dataset?
  - I've never seen anyone try to seriously use an LLM to solve instances of the TSP. Why should an LLM be good at it?
  - GSM8k is well known to be highly contaminated, see "GSM-Symbolic: Understanding the Limitations of Mathematical Reasoning in Large Language Models" by Mirzadeh.
  - I'm unclear about some pretty basic aspects of the analysis: e.g. what LLM did you use? I'm expecting to see paragraphs of details about Llama versions, context length, GPUs, quantization, etc. but none of this is present.
  - I'm expecting more notions of goodness than just F1 score.

Would like to see a link to code.
Would like to see the language toned down, e.g. proclaiming one's own work "pioneering" is not helpful.
The title is vague.


Some typos: (1) Table 3: "Strategey", (2) "heavy hand" is not idiomatic English, I guess you mean "heavy handed"?

**Strengths And Weaknesses:**

Positives
 - Gives an interesting view of LLM applications. As someone who works every day on small details of LLMs, it's interesting to see a novel perspective.

Negatives
 - Put simply: having some novel perspective does not make a paper. IMO it needs to have either (1) a strong formal basis, or else (2) compelling experimental validation. The diagrams attempting to elucidate the connections are not (1), and the limited and detail-light experiments are not (2).

---

### Review · Reviewer_HRR8 · 2025-01-03

**Summary Of Contributions:**

The paper proposes a novel framework for constructing Large Language Model (LLM) applications in a modular fashion, drawing an analogy to the architecture of deep neural networks. The authors suggest that different LLM components, such as retrievals, memories, and prompting strategies, mirror the modular components of deep learning models like MLPs, attention mechanisms, and recurrent modules. They introduce forward inference and feedback mechanisms for LLMs. In this framework, prompts are treated as weights, and prompt optimization from feedback is likened to the backpropagation algorithm used in deep learning. The paper also introduces an algorithm for optimizing the architecture of LLM applications, similar to NAS in deep learning research, resulting in an 11% performance improvement over randomly designed architectures.

**Audience:**

No

**Broader Impact Concerns:**

No questions or concerns.

**Claims And Evidence:**

No

**Requested Changes:**

* Provide a more rigorous definition of the "translation" between deep learning and LLM building blocks. This should include specific criteria and potentially empirical justification for the proposed analogies. The authors need to clearly outline the parameters and limitations of these analogies to establish their validity. For example, justify the use of the term "Transformer" in Section 3.3. Explain what aspects of the design warrant this label, given the established understanding of Transformer architectures in deep learning.
* Clarify the process of updating the meta-prompt in the feedback propagation mechanism. Provide a detailed description of how generated reflections and responses are used to modify the meta-prompt, ensuring that the methodology is reproducible and understandable.
* Expand on the novelty of the models presented. If the models themselves are not new, emphasize the novel aspects of their application within the proposed framework. Consider incorporating more diverse or complex building blocks to demonstrate the versatility of the approach.

**Strengths And Weaknesses:**

Strengths:
* The analogy between deep learning architectures and LLM application design is curious.
* The authors provide interesting empirical results, showing performance gains from modular design, prompt optimization, and architecture search.

Weaknesses:
* The paper reads more like a position paper than a rigorous ML paper, with a focus on high-level concepts rather than detailed technical contributions.
* The paper needs to establish more rigorous criteria for the "translation" between deep learning and LLM building blocks. The conceptual analogies, while intriguing, lack a formal or empirical basis, for examples like equating Memory Processing with RNNs without considering the sequential nature of RNNs and the absence of a hidden state in LLM memory processing.
* The models described in Section 4.2 are not novel, raising questions about the originality of the technical contributions beyond the conceptual framework. The model presented in section 3.3 lacks clear justification of why it should be called a transformer.
* Certain aspects of the implementation, such as the process of updating the meta-prompt based on generated reflections and responses, are not clearly explained.

---

### Decision · Action_Editor_o95e · 2025-02-09

**Recommendation:** Reject

**Comment:**

This paper proposes a modular framework to optimize LLM applications by drawing analogies between LLM building blocks and the successful deep learning modules. This framework includes techniques including prompt optimization through forward inference and feedback mechanisms, a search algorithm for the best configuration of LLM workflows.

Reviewers acknowledge several strengths of this paper. It is a novel and interesting perspective to view LLM applications as modular, optimizable workflows rather than a hand-engineered tuning process.

However, there are several major weaknesses in both formal guarantees and experimental results. The mapping between LLM components and deep learning architectures is arbitrary and lacks more rigorous formalization and deeper rationales. More justification would be needed to explain this analogy and architectural choices. The experiment is not comprehensive because the chosen dataset is narrow, and only 30 cases of GMS8k are used for evaluation. More challenging datasets such as MMLU, MATH, GPQA are required.

In addition, reviewers have also provided a myriad of suggestions and questions for writing improvements, and the authors are encouraged to revise their draft accordingly.

To sum, while it is a promising work, the current manuscript is still not ready for publication. Authors are strongly encouraged to revise the paper given the reviewers' feedback.

**Audience:**

Yes

**Claims And Evidence:**

No

**Resubmission Of Major Revision:**

The authors may consider submitting a major revision at a later time.